# A Study of a Specialised American Police Discourse Genre: Probable Cause Affidavits

Audrey Cartron 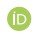

Department of Applied Foreign Languages, Faculty of Foreign Languages and Cultures, Nantes University, CRINI, 44000 Nantes, France; audrey.cartron@univ-nantes.fr

**Abstract:** This paper focuses on the analysis of a specialised American police discourse genre and is based on a corpus of 115 probable cause affidavits. A probable cause affidavit is a sworn statement written by American police officers to state that there is probable cause to believe the defendant has committed (or is committing) a criminal offence and that legal action is required. After briefly presenting the methodological framework for this study, the paper intends to show how the police use specific linguistic, discursive and rhetorical strategies to serve a specialised purpose, which is to present the existence of probable cause to the relevant legal authorities. The findings indicate that officers use various discursive devices to inform but also—and perhaps more importantly—to convince their audience by means of a chronological and structured narrative of events that follows a prototypical three-fold internal organisation (exposition, investigation, resolution) signalled by specific linguistic markers. Finally, the paper intends to go beyond the objective description of events in order to highlight the assertive nature of this discourse genre and the additional rhetorical strategies used by PCA writers. It studies the emphasis placed on the expertise of the author, as well as the police classification of the offence and the progressive elaboration of the burden of proof.

**Keywords:** corpus linguistics; discourse analysis; English for Police Purposes; English for Specific Purposes; genre analysis; move analysis; probable cause affidavit; specialised discourse

## 1. Introduction

Due to the multiple interactions between police forces (specialists) and other members of society (non-specialists), English for Police Purposes (EPP) might intuitively appear less specialised (Petit 2010, §12) than Scientific English, for example. Nevertheless, EPP can be considered to be a specialised variety of English located at the crossroads of forensic and legal languages, with specific linguistic (Philbin 1996; Poteet and Poteet 2000), discursive (Johnson et al. 1993; Gaines 2011; Rock 2017) and cultural (Fielding 1994; Reiner 2000; Cartron 2023b) characteristics that deserve to be studied in depth. Among the various approaches that can be used to investigate specialised languages, genre analysis provides an interesting insight into the specialisation of the discursive community and its practices, taking into account both linguistic and extralinguistic features (Swales 1990, pp. 24–27; Beacco 2004, p. 116; Bhatia 2017, p. 6). As far as English for Police Purposes is concerned, this specialised variety of English is characterised by a diversity of genres, both spoken—such as police interviews, radio communications or court testimonies—and written—police reports, manuals or codes of ethics, for instance[1].

This paper focuses on the analysis of a specialised American police discourse genre belonging to the category of police reports and is based on a corpus of 115 probable cause affidavits (PCAs)[2] written by American police officers from different police forces (police departments, sheriff and county law enforcement agencies, as well as federal law enforcement agencies). In the United States, police officers are required by the Fourth Amendment of the Constitution to present probable cause and to justify that legal action is required:

> The right of the people to be secure in their persons, houses, papers, and effects, against unreasonable searches and seizures, shall not be violated, and no Warrants shall issue, but upon probable cause, supported by Oath or affirmation, and particularly describing the place to be searched, and the persons or things to be seized (Library of Congress n.d.).

In order to do so, officers write a probable cause affidavit[3], a sworn statement to state that there is probable cause to believe the defendant has committed (or is committing) a criminal offence and that the facts support the claim to make an arrest, conduct a search or seize the property (Crespo 2020, pp. 1279–80). Three different degrees of proof can be identified in the American legal system: reasonable suspicion, probable cause, and beyond reasonable doubt. Probable cause is the intermediate burden of proof and requires more evidence than reasonable suspicion (Taslitz 2010, p. 146) but less than beyond reasonable doubt. Therefore, it is an intermediate burden of proof between suspicion and certainty, and the police must gather sufficient evidence—both qualitatively and quantitatively—to support the hypothesis of the respondent's guilt.

Probable cause affidavits provide a brief summary of the events and identify the main parties involved, such as the victim(s), suspect(s) and witness(es). PCAs form a set of textual productions with a single communicative aim: to present the facts objectively to legal authorities (police superior, district attorney, judge or other actors in the judicial process). Based on the police officers' statements—as well as on other evidence and information from the case—the competent judicial authorities can validate (or not) the existence of probable cause; that is to say, they can determine whether there are grounds to believe the defendant has committed (or is committing) a criminal offence. Probable cause affidavits can be drawn up in several situations: either the individual has already been taken into custody, and the police must show the judge that probable cause exists to justify the legal value of the arrest, or this has not yet happened and the police must prove probable cause in order to ask a judge to issue an arrest warrant. The document can also be written as part of an application to a judge for a search warrant.

In the literature dealing with English for Police Purposes, several lines of enquiry relating to the discursive practices of police officers can be identified. Studies on police discourse tend to focus on specialised communication and practices as well as on major police discourse genres. They mainly deal with suspect interviews (Baldwin 1993; Leo 1996; Magid 2001; Haworth 2006; Benneworth 2009; Cartron 2023a), victim/witness interviews (Rock 2001; Milne and Bull 2006; Dando et al. 2009), police reports (Coulthard 2002), police calls (Tracy and Tracy 1998; Rock 2018), caution and Miranda warnings (Rock 2007; Heydon 2013), radio communications (Glaister 2006), interactions with professionals of related specialised fields (Johnson 2003; Charman 2013), police humour (Holdaway 1988; Gayadeen and Phillips 2016; Cartron 2023b), and *policespeak* (Fox 1993; Johnson et al. 1993; Hall 2008). However, according to the author's knowledge, no extensive and in-depth linguistic and discourse analysis has been conducted on the specialised American genre of probable cause affidavits.

After describing the methodological framework on which this study is based (Section 2), the paper provides a detailed move analysis of probable cause affidavits and shows how police officers introduce the existence of probable cause to the relevant legal authorities through the presentation of a chronological and structured narrative of the events (Section 3). The article then presents additional rhetorical strategies used by PCA authors and sheds light on the probative value of this discourse genre. It intends to go beyond the objective description of facts in order to highlight the emphasis placed on the expertise and reliability of the author, as well as the underlying progressive elaboration of the burden of proof (Section 4).

## 2. Materials and Methods

### 2.1. General Methodological Framework and Research Question

Several authors (Petit 2002; Wozniak 2011; Van der Yeught 2016; Stark 2020; Cartron 2022) have contributed to the development of a tripartite methodological protocol aimed at proposing descriptive characterisations of specialised varieties of English (SVEs). This three-fold protocol is based on the study of the discursive, linguistic and cultural features of the specialised domain under study. These three approaches are complementary and offer the possibility to present a holistic, structured and methodological description of specialised languages. The present study focuses on a discursive approach to English for Police Purposes and takes into account the linguistic and extralinguistic characteristics of the specialised language under study, including the context of production, the identity of the actors participating in the communicative event, the specific features of the specialised field as well as the aims of the communicative event (Charaudeau 2009, p. 41). Vijay Bhatia (1993, pp. 22–35) stresses the need to (re)place a discourse genre in situation and in context. A probable cause affidavit, for instance, cannot be analysed without taking into account the context in which it is produced, whether it is the immediate textual context (peritext), the context of reception (which may be reflected in the presence of the author and the addressee in the text, for example), the context of production (a particular offence, involving specific actors, at a given moment), or the social and cultural context, and more specifically the American judicial context (the concept of probable cause as specific to the United States, the legislation in effect in the state where the alleged offence was committed, etc.).

This paper deals with the detailed study of probable cause affidavits. Following the pioneering work of John Swales (1990), the rhetorical organisational patterns of PCAs were studied through a detailed move analysis of the genre. This approach consists of identifying the discursive or rhetorical units (called "moves") that perform a specific communicative function and serve the overall specialised purpose of the genre. In order to analyse a representative sample of the genre under study, it was decided to gather a corpus of authentic productions from American police officers. Two different approaches can be considered to investigate a corpus:

> "corpus-based" investigations, which are undertaken to check the researcher's intuition about language use, and "corpus-driven" investigations, where the researcher approaches the corpus data with an open mind to see what patterns emerge (Nesi 2013, p. 407).

The present study focuses on corpus-based investigations and concentrates on the following research question: how do police officers use specific discursive, linguistic—in terms of lexicon, phraseology and syntax—and rhetorical strategies in probable cause affidavits to serve a specialised purpose, which is to present the existence of probable cause to competent legal authorities? However, as it would be reductive to be limited by the rigid framework of a starting hypothesis (Martin 1997, §18), the author remained open to other leads or significant aspects that might emerge from the corpus during its exploration.

### 2.2. Overcoming the Lack of Accessibility of Sources

To study the specificities of EPP genres, it is necessary to gather authentic productions from police officers in order to undertake detailed and targeted analyses of specialised discourse. Indeed, the study of the discourses emitted by a given specialised community must be based on corpora composed of primary and authentic sources (Wozniak 2019, p. 5). However, in the field of EPP, collecting authentic materials—written and oral—produced by professionals is not an easy task (Oxburgh et al. 2010, p. 59). Internal productions within the professional police community can be confidential in order to guarantee the presumption of innocence[4], to protect victims and witnesses, to ensure the safety of police officers and their families and to avoid any effect on ongoing investigations. As Brodeur and Monjardet (2003, pp. 11–12) point out, in many countries, the legitimacy of police

secrecy is sanctioned by law. In the United Kingdom, for instance, data protection laws prevent many police documents from being made accessible to the general public.

In the United States, the Freedom of Information Act (1966) defends the principle of the right to information and makes it mandatory for federal agencies to hand over their documents to anyone who requests them. However, the legal obligation to make police documents accessible or not depends on the legislation in each state. In some states, journalists specialised in criminal cases, legal professionals, or even ordinary citizens can send requests for access to files on past or current cases. The Berkeley Graduate School of Journalism addresses the complex question of access to police documents in the United States and provides an online guide listing several sources that make authentic police records available to the public, including the American website *The Smoking Gun* (Grabowicz 2014). Created in 1997, this website belongs to the American group Turner Broadcasting System, a subsidiary of Warner Media, which runs, among others, the news channel *CNN*. The website is specialised in the publication of legal documents (Carr 2008), including police reports, arrest records and probable cause affidavits obtained through different sources: from government and law enforcement sources, via Freedom of Information requests, and from court files nationwide (The Smoking Gun 2020). Journalists of *The Smoking Gun* investigate criminal offences, and they publish police or court documents relating to these cases on the website[5]. PCAs studied in the present paper were selected from the *Smoking Gun* website.

### *2.3. Collecting and Analysing Data*

Since the objective was not to study the diachronic evolution of probable cause affidavits over time, a synchronic perspective was adopted, and the collection of documents was limited to three years. Texts published on *The Smoking Gun* website between January 2018 and December 2020 were pre-selected. Among the 622 documents that were published, only those belonging to the category of probable cause affidavits and exclusively those for which it was possible to identify the date when it was written, as well as the author (or at least the corresponding police force), were included. Documents that were incomplete (missing pages) or unofficial (labelled "Unofficial document", "Unofficial copy", or "Not certified copy") were discarded. As the files available on the website were scanned versions of original documents, they were in image format (.jpeg files). They were then converted to text files (thanks to an optical character recognition software[6]) so that computerised analyses could be carried out using the corpus analysis toolkit AntConc (version 3.5.8.0). In order to make sure that the original and the converted texts were identical, each document was carefully proofread to correct the numerous missing, misspelt or truncated words and other typographical, linguistic or punctuation errors that were generated during conversion. Some texts were also entered manually when the conversion tool did not provide a usable result. These different steps led to the constitution of a corpus of 68,133 words, gathering 115 probable cause affidavits from 68 different American law enforcement agencies and from 18 different states. Although constrained by the question of the accessibility of the sources, the size of the selected corpus seemed adequate to study the process of specialisation at work in this specialised genre and to carry out quantitative and qualitative analyses jointly.

In order to study the multi-faceted genre of PCAs, a modular approach (Roulet n.d., p. 21) has been chosen based on the idea that a genre can be considered as a system combining various smaller parts called "modules". The lexical module looks at lexical units (nouns, adverbs, verbs, adjectives, pronouns) and the use of specific vocabulary or terms. The phraseological and syntactic module studies collocations, fixed phraseological units and, more generally, the relations between linguistic units (the use of active and passive voices or indirect discourse, for example). The structural module covers the formal characteristics of the genre, its internal structure (rhetorical moves) and its external structure (paratext). The combination of these three main modules leads to the accumulation of

knowledge on the specificities of the genre and, beyond that, of the specialised variety being studied.

Several tools and methods were used to study the various elements within each module. Firstly, careful reading and manual analyses of the selected texts were carried out throughout the process of collecting the corpus. Specific attention was paid to the lexical, phraseological, syntactic and structural characteristics of the genre. Detailed move analyses were also performed by the author on five probable cause affidavits from different US states and types of police forces. The procedure used to study discourse moves in PCAs included understanding the overall rhetorical purpose of the texts, identifying the different text segments as well as their function and purpose, and then studying and coding common functional and/or semantic themes represented by the various text segments (Kanoksilapatham 2007, p. 33). Secondly, this first-hand qualitative approach was supplemented by more quantitative and computerised processing of the data (Banks 2016, §33) to complete the characterisation of this discourse genre. As underlined by Budsaba Kanoksilapatham, a corpus-based analysis allows "for more complex and generalizable research findings, revealing linguistic patterns and frequency information that would otherwise be too labor intensive to uncover by hand" (Kanoksilapatham 2007, p. 36). For this study, the AntConc concordance was chosen because it offers the possibility of easily studying the behaviour of a word in context (*keywords in context* feature), as well as its distribution and place in each text of the corpus (*concordance plot*). It also allows us to identify the most frequently used words in the corpus (*word list*) and to single out collocations or compound terms (*clusters/n-grams*). Finally, some authors also advocate an ethnographic approach to the genre, which involves the validation—or invalidation—of analyses by a specialist in the field (Bhatia 1993, pp. 22–35). The genre of probable cause affidavits—including their content, aim, and purpose—was thus discussed with American (as well as some British) police officers who were interviewed between December 2019 and March 2022. The following sections of the article describe the findings of this study on the characterisation of probable cause affidavits.

## 3. Move Analysis of a Chronological and Structured Narrative of Events

### 3.1. A Three-Fold and Prototypical Internal Structure

The internal structure of probable cause affidavits is not fixed, but several regularities emerge. Following John Swales's genre analysis approach (Swales 1990), the results of the present research indicate that PCAs generally adopt a prototypical structure characterised by three rhetorical moves signalled by specific linguistic markers. It is organised around a chronological narrative of the facts, and the three moves follow a prototypical narrative structure in three acts: (1) exposition, (2) investigation, and (3) resolution/denouement. The first move is devoted to the presentation of the initial context. After specifying the exact date, time and location of the intervention, the police officer presents the triggering event or inciting incident (emergency call, *flagrante delicto* observed during a patrol, transfer of a file between two police units or forces . . .) and the type of offence under investigation. In the second move, the police officer sheds light on the various investigative steps taken (such as taking statements or viewing recordings from the cameras that filmed the scene) and the evidence collected. In the third and final move, the conclusion is made up of the findings on the existence of probable cause and details of the arrest of the suspect when applicable. Therefore, the reader is guided step by step by the author, who presents the events in a chronological and structured manner. The breakdown of affidavit PC_LA_WestMonroePD_2019[7] (Figure 1) illustrates the use of this prototypical three-stage structure[8].

The actions of the police are not always explicitly mentioned. Some affidavits present a chronological narrative of the offence itself, with an omniscient point of view and a focus on the actions of the suspect rather than on the investigation. However, the elements presented in the document are similar: initial context, inciting incident, implicit presentation

of the investigation, and collection of evidence. In PCAs, each move has specific linguistic characteristics, as exemplified in the following subsections.

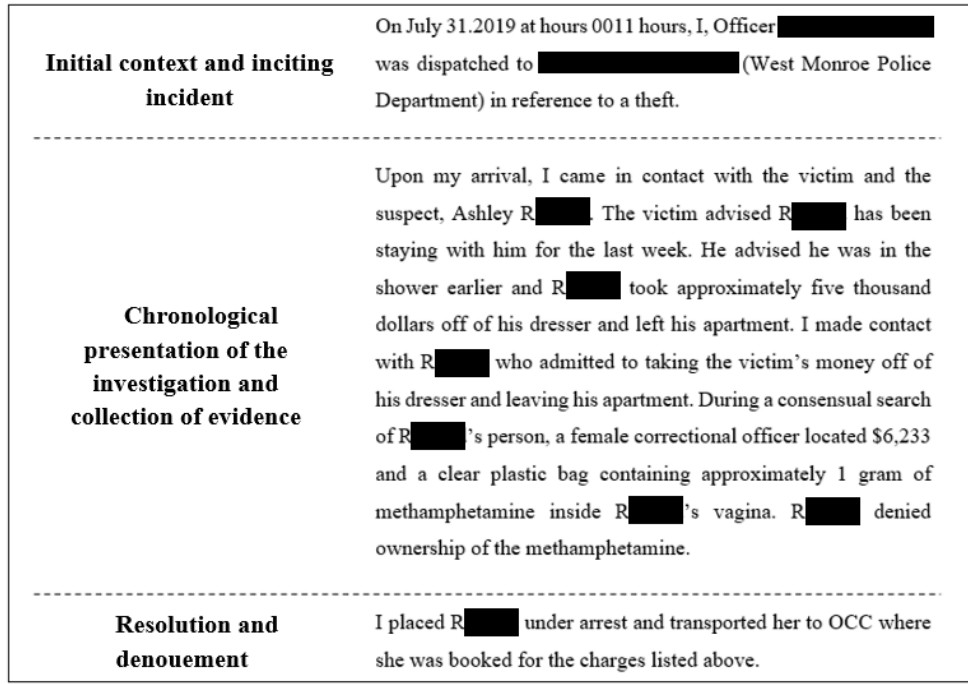

**Figure 1.** Prototypical structure of probable cause affidavits: example and breakdown of affidavit PC_LA_WestMonroePD_2019.

*3.2. Examples of Linguistic Markers for Move 1 (Exposition)*

The first words of probable cause affidavits always set the facts in a precise temporal and geographical context. The dates, times and places of police intervention are the first elements mentioned, generally followed by the type of offence, plunging the reader *in medias res* in the recounted events. This pattern is recurrent in police reports, as an American police officer underlined:

> There definitely is [a police-style of writing]. When it comes to police officers or Detectives writing reports, sure, it's a definite style. It's very mechanical. There isn't a lot of fluff. It usually starts out on the day, date and time. So, "On Thursday, May 4th, at about eleven ten a.m., myself, Sergeant [*states his own name and surname*], on Squad 21 15 observed . . .", then you go into whatever the story is (date of the interview: 4 June 2020).

Move 1 of PCAs is characterised by the extensive use of contextual linguistic units (adverbs, prepositions, prepositional phrases, verbs, etc.). For instance, the locations and types of incidents are presented with specific recurring linguistic markers (Table 1).

**Table 1.** Recurrent linguistic markers in the first rhetorical move of probable cause affidavits.

| Linguistic Markers Introducing the Location of the Intervention/Incident | Linguistic Markers Introducing the Type of Incident |
|---|---|
| *responded to* (40 occurrences) | *for/on a report of* (11 occurrences) |
| *was/were dispatched to* (18) | *in reference to* (32) |
| | *responded to* (3) |
| | *was/were assigned to* (2) |
| | *was/were dispatched to* (2) |

The contexts of the use of these markers and their distribution in PCAs indicate that they form a linguistic specificity of the first rhetorical move. For instance, Figure 2 is

a screenshot of the Concordance plot tool of AntConc, showing the distribution of the prepositional phrase *in reference to* (32 occurrences) in different texts[9]. It demonstrates that the item is used extensively and exclusively at the beginning of affidavits to introduce the type of offence.

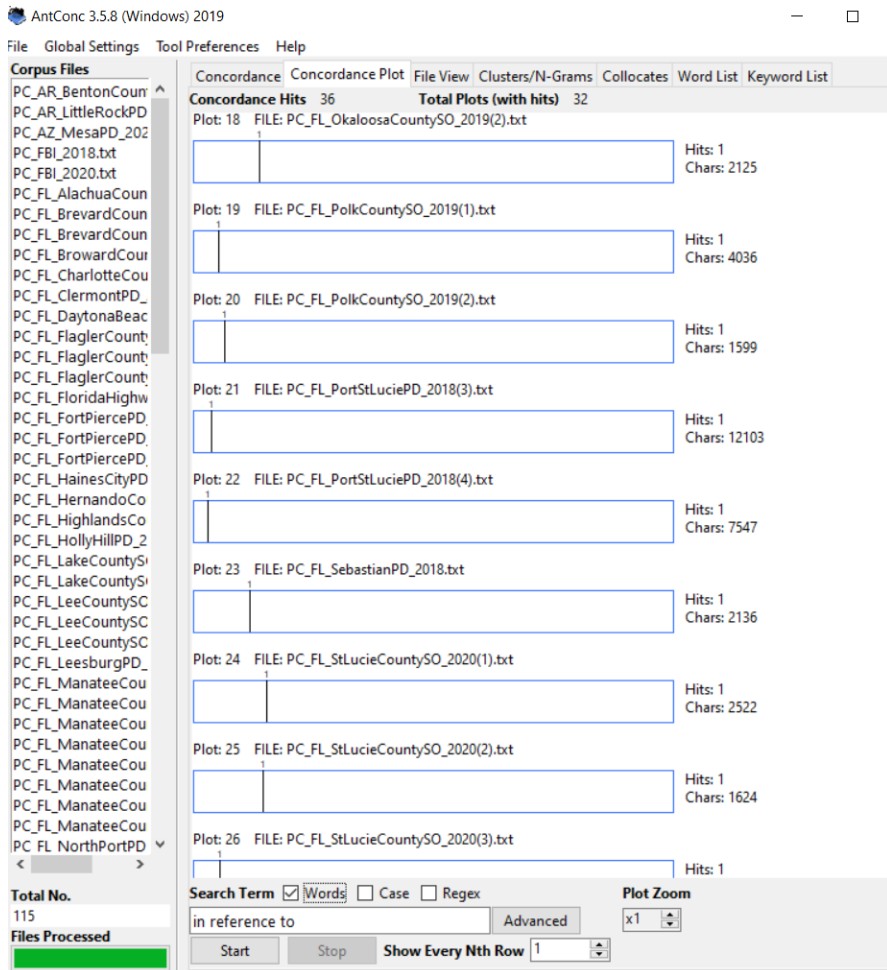

**Figure 2.** Distribution of the prepositional phrase *in reference to* in probable cause affidavits (AntConc, Concordance plot).

Furthermore, Figure 3 illustrates the contexts of use of this prepositional phrase and shows that it collocates with nouns designating generic categories of incidents. These nouns may be legal terms given to the offence in the law (*theft*, *battery*) or, in most cases, a much vaguer classification (*disturbance*, *sexual offence*, and even *suspicious incident*).

Interestingly, this initial description of the offence reflects the information given to the police officer when they are assigned to the case and reveals their initial imprecise knowledge of the facts when they are dispatched. The investigation then enables the classification of the criminal offence more precisely, as it will be discussed in Section 4.2.

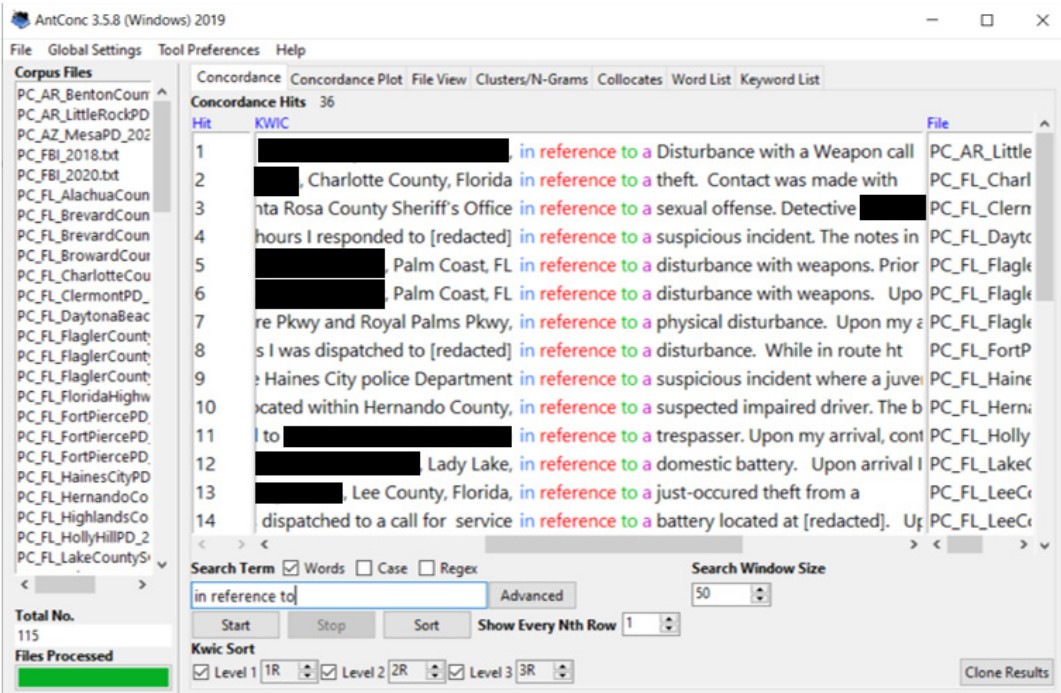

**Figure 3.** Concordance lines for *in reference to* in PCAs (AntConc, KeyWord In Context).

*3.3. Examples of Linguistic Markers for Move 2 (Investigation)*

In the second move, the different steps of the investigation are precisely traced using various temporal markers such as *after*, *before*, *during*, *hours*, *later*, *then*, *time*, *when* or *while*. It shows the need for a very thorough description of the facts. The frequent use of *approximately* (161 occurrences) was deemed intriguing as it seemed to contradict the emphasis on precision specific to the writing of probable cause affidavits. However, a study of the contexts in which this adverb of approximation was used revealed that it often collocates with extremely specific temporal details, such as *at approximately 0124 h*, paradoxically reinforcing exhaustiveness.

Moreover, law enforcement representatives talk to a wide range of people: victims, suspects, witnesses, and other specialists (forensic experts, police colleagues present at the crime scene or previously in charge of the case). Each protagonist is clearly identified using categorising nouns in order to establish the agents of the various actions reported. The significant use of *defendant* and *victim* can be highlighted, as they are the first two most frequently used common nouns in the corpus, with 460 occurrences (rank 18) and 369 occurrences (rank 22), respectively. The regular use of the nouns *officer(s)* (196 times), *deputy* (118), *police* (99), and *affiant* (69) can also be highlighted. Finally, third-person pronouns are also numerous, and *he*, *him*, *his*, *she* and *her* are among the twenty most frequent words in the corpus. Additionally, investigative acts carried out by the police are also clearly identified and indicated by the use of verbs such as *observed* (139 occurrences), *asked* (122 occurrences), *made contact with* (42 occurrences) or *spoke to/with* (43 occurrences). To this extent, probable cause affidavits provide insight into the practices of the specialised community. For example, the following extract from an affidavit drafted by a Florida police officer illustrates the procedure to be followed in the event of suspected drunk driving:

> Deputy S arrived on scene and assisted with demonstrating the Standardized Field Sobriety Exercises. Deputy S explained the horizontal gaze nystagmus exercise to the defendant and he replied he understood the instructions given. [. . .] Deputy S asked him multiple times to only follow the tip of the pen with his eyes and reminded him not to move his head. The defendant continued to move his head [. . .]. Deputy S then explained and demonstrated the walk and turn exercise to the defendant. The defendant was unable to stand in the

heel to toe position without losing his balance [. . .]. Deputy S then explained and demonstrated the one leg stand to the defendant. [. . .] The defendant then stood with his feet next to each other without lifting a foot up. The defendant was reminded to pick a foot of his choosing to complete the exercise. [. . .] The defendant raised his foot for approximately half of a second before losing his balance and setting his foot down. [. . .] After my investigation I determined the defendant was under the influence of an alcoholic beverage and operating his golf cart under the influence of alcohol (PC_FL_SumterCountySO_2020(1)).

Therefore, PCAs depict—whether explicitly or implicitly—the gestures, practices and procedures and the day-to-day life of an American policeman in the field.

Last but not least, the extensive use of indirect discourse and reported speech verbs can also be highlighted. Among the hundred most frequently used words of the corpus, the following verbs were identified: *stated* (rank 20, 420 occurrences), *advised* (rank 42, 178 occurrences), *said* (rank 51, 155 occurrences), and *told* (rank 63, 123 occurrences). The preterit *stated* is used frequently; it is the twentieth most used word and the second most used verb (after *was*) in the corpus. It appears in 82 of the 115 texts and is frequently used several times in the same document, as shown in Figure 4. This recurrence can be explained by the fact that the derivative *statement* refers to words declared before a police officer and intended to be produced in court.

The occurrences *stated* are spread throughout probable cause affidavits and signal the use of reported speech and the presentation of information obtained during the various interviews conducted during the investigative work. The syntactic rule provides for the adaptation of pronouns when using indirect discourse, but some errors were identified in the corpus, remains of an incomplete transition from direct to indirect speech, as in the following example:

While sitting in the turning lane on Highway 27, the defendant told the victim to get out. The defendant stated the police will find *you* a new home (our italics, PC_FL_HainesCityPD_2019).

Interestingly, despite the wide variety of words belonging to the class of declarative verbs, *state*, *advise*, *say* and *tell* are selected as priorities by affidavit writers. This lexical preference for a restricted spectrum of verbs is corroborated by the few occurrences of verbs with similar semantic characteristics. For example, the verbs *added* (1 occurrence as a verb of declaration introducing reported speech), *explained* (20 occurrences), *indicated* (7 occurrences), *mentioned* (1 occurrence) or *reported* (10 occurrences) are very rarely used. Other variants are never used in the corpus, such as the verbs *declared*, *highlighted*, *underlined* or *pointed out*. This lack of variation in the formulations seems to indicate that the facts presented in PCAs take precedence over the form, as the writing is mostly motivated by a concern for concision, brevity, clarity and efficiency. Moreover, this low vocabulary richness also suggests that PCAs are very formulaic in nature. This genre is frequently written by police officers, and, as a result, lexical choices, as well as collocations, become fixed and routinised.

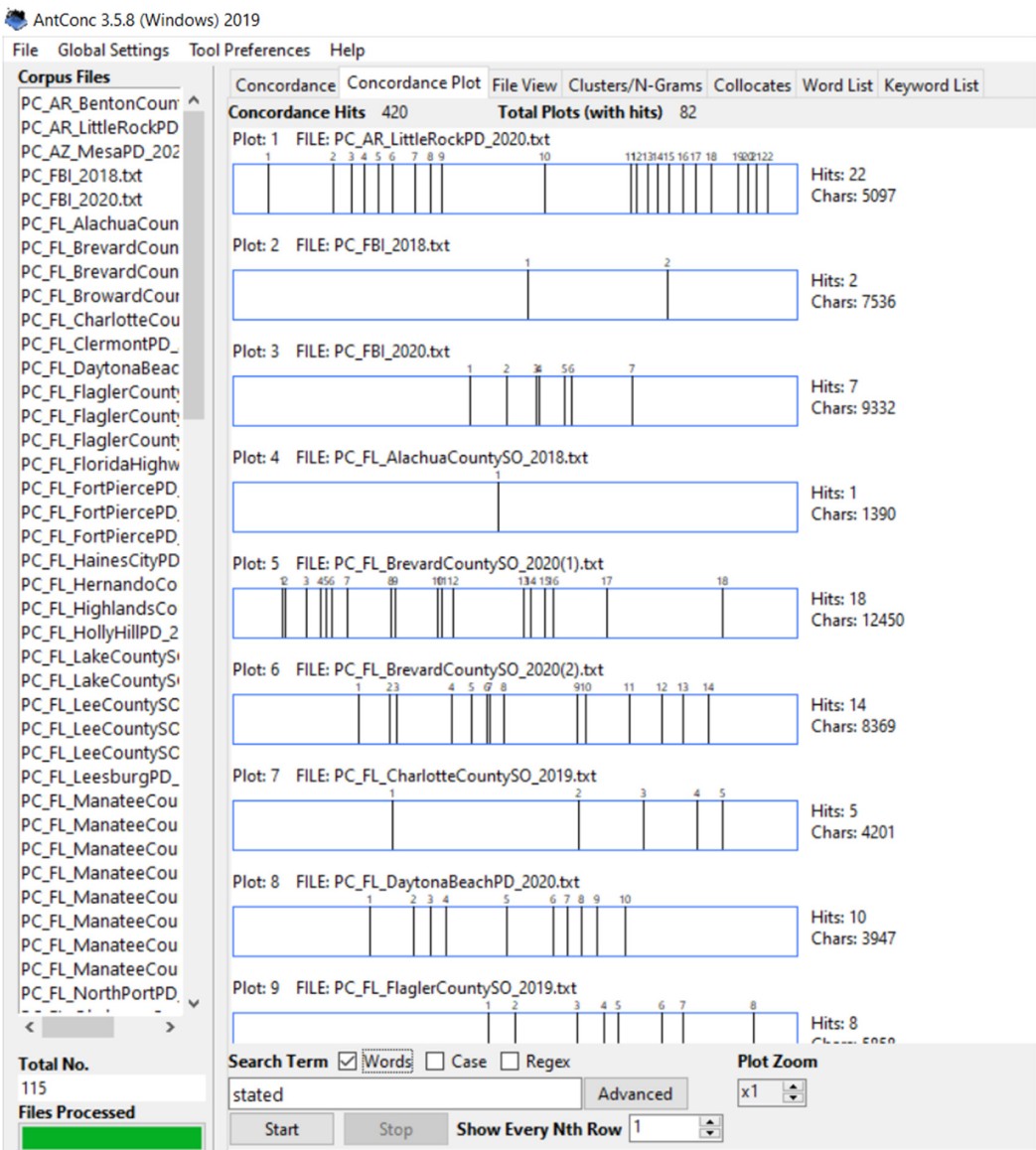

**Figure 4.** Distribution of *stated* in PCAs (AntConc, Concordance plot).

*3.4. Examples of Linguistic Markers for Move 3 (Resolution)*

Move 3 concludes probable cause affidavits and presents, in a few words, the police officer's conclusions following the investigation they have conducted. The concluding elements differ from one police force to another, and there are many variations in this rhetorical move. In some documents, the author stresses that the evidence gathered establishes the existence of probable cause. The officer indicates that all the elements are present to observe a breach of the law and precisely designates the offence(s) committed and the corresponding legal text(s). Certain lexical elements are specific to this rhetorical move. The pattern <*Based on* [evidence], *probable cause* . . .> is used several times (25 occurrences), as in the following example:

> *Based on* the above facts, statements and physical evidence provided, *your Affiant has probable cause to believe* and does believe that the above listed probable cause, all lead to the substantiation that defendant, S, has committed a violation of the laws of the State of Florida, to wit: Solicitation to commit 1st degree Murder, contrary to section 777.04 (4-B), Florida Statutes and Solicitation to commit an Occupied Burglary with a Battery, contrary to section 777.04 (4-C) (our italics, PC_FL_BrevardCountySO_2020(1)).

The evidence referred to in the conclusion ("the above facts, statements and physical evidence") are relatively vague categories with anaphoric value, as they refer to the evidence previously referred to. In addition, at the end of the affidavit, some authors highlight the actions taken by the police to close the case, and more specifically, the arrest of the respondent, as shown by the last words of this probable cause affidavit:

> Based on my observations on scene, *I took M into custody* for FSS 784.045(1A1)—Aggravated battery for striking the victim on the head with the can of Spaghetti's. *M was transported to St Lucie County Jail without incident. This case was Cleared by Arrest* (our italics, PC_FL_StLucieCountySO_2020(2)).

Therefore, probable cause affidavits are chronological narratives of the facts, structured in three stages, and each move of this prototypical structure serves an overarching communicative purpose (Bhatia 1993, p. 37): to inform and guide the reader but also to convince legal authorities of the existence of probable cause. In the course of this study of probable cause affidavits, it became clear that the communicative aim of PCAs is not only to present a series of facts objectively but also to model the discourse in order to serve a specialised purpose and, more broadly, to provide usable content for the judicial process. The last section of this article argues that police officers use specific discursive procedures to inform, but also—and perhaps above all—to convince and persuade the reader(s) of the guilt of the individual, and not just of its probability.

## 4. Additional Rhetorical Strategies: From Probability to Certainty?

### 4.1. The Author's Expertise and Credibility

In police reports, discourse modelling is motivated by the underlying desire to convince the reader of the veracity of the facts presented. As in academic genres studied by Ken Hyland (2005, pp. 173–74), police reports are written with the aim of persuading the reader by using various rhetorical techniques, including the credible representation of themselves, their actions and the events observed:

> [A]cademics [are] not simply producing texts that plausibly represent an external reality, but also as using language to acknowledge, construct and negotiate social relations. Writers seek to offer a credible representation of themselves and their work [and] controlling the level of personality in a text becomes central to building a convincing argument. Put succinctly, every successful academic text displays the writer's awareness of both its readers and its consequences (Hyland 2005, pp. 173–74).

In PCAs, the expertise of the author is sometimes explicitly presented. In some police forces, affidavits begin with an introductory paragraph that briefly describes the officer's career: number of years of service, skills acquired during various training courses, types of cases handled, etc. In the State of California, this introductory paragraph is informally referred to as the "hero sheet":

> The way that we write our affidavits in the State of California usually starts with what we jokingly refer to as the hero sheet. We explain to the judge who we are, and when we're forming our affidavit, we refer to ourselves, the person that is swearing to the facts and circumstances that we're in this affidavit, as we are seeking this search warrant. We refer to ourselves as the affiant, or sometimes people will pronounce it as affiant. So, in that hero sheet section of the affidavit at the beginning I explain my training and experience, because later on in the affidavit I'm going to ask the judge to take my expert opinion into account when I sum up the meaning of all those facts and circumstances, and what they mean as I lay out the basis for probable cause (Richardson 2018).

Several examples of this explicit presentation of the author's expert status were found in the PCA corpus, as shown by the following extracts from a police officer in North Dakota and from an FBI agent:

I Detective J, attest to the following: That *I am a trained and licensed Peace Officer* with *9 years of experience* with jurisdiction to enforce state law in city of Bismarck, Burleigh County, North Dakota. In 2009, *I successfully completed Military Police Academy for the United States Marine Corps* in Fort Leonard Wood, MO. In 2010, *I attended the Devils Lake Regional Police Academy* and was hired by the Mandan Police Department in 2010. In 2013, I was hired by Bismarck Police Department and currently work as an Investigator in the Investigation Section. *I have attended The Basic Course of Criminal Investigation by BCI, The Reid Investigator Interview and Advance Interrogation and Evidence Based Interrogation by the CTK Group. I have attended the National Fire Academy and taken Fire Investigation Essentials to Origin and Cause.* I have *over 1300 h of Law Enforcement related training* (our italics, PC_ND_BismarckPD_2019).

I am *a Special Agent with the Federal Bureau of Investigation* (FBI) within the United States Department of Justice and have been so employed since March 2000. I *primarily work in the Minneapolis, Minnesota division.* Prior to my employment with the FBI, *I served as an Indiana State Trooper for approximately 3 years.* As a Trooper *my duties included criminal investigation, traffic offenses, and gaming regulation.* During my tenure with the FBI, *I have actively participated in investigations, including violent crimes* in Indian County *and international terrorism.* Since 2009, I have been *the Minneapolis Division Weapons of Mass Destruction Coordinator* and have *experience investigating explosives.* I have *a Bachelor's Degree* from Indiana University (our italics, PC_FBI_2020).

Various elements are mentioned, including the author's status, training institution(s), number of years of experience, previous places of practice, current assignment and various training courses received. This accumulation actively contributes to the construction of the author as a credible expert in the field of law enforcement. To a certain extent, the facts relating to the offence subsequently stated are difficult to contest because they are backed up by credentials and in-depth professional expertise. In PCAs, police officers elaborate and structure narratives by utilising the "rhetorics of reality" and, more specifically, the "reality production kit" evoked by Alexa Hepburn (2003, p. 181). For instance, the authors foster category entitlement by "construct[ing] [their] talk as coming from a category that is credible or knowledgeable in a way that is relevant to the claim" (ibid.). Additionally, expertise is recognised by the courts as they rely on the training and experience of police officers to assess probable cause:

> [T]he [Supreme] Court has been reasonably consistent in explicitly stating, or at least assuming, that a police officer's training and experience help support the existence of probable cause and reasonable suspicion. And the lower courts have followed suit (Kinports 2010, pp. 752–54).

In order to establish the existence of probable cause, police officers must rely on their expertise and knowledge regarding legal definitions of offences, well-known local criminal characters, different types of *modus operandi*, various investigative approaches and interview techniques (South Carolina Law Enforcement ETV Training Program 1976b, pp. 19–20). Thanks to their specialised knowledge, police officers can, for example, interpret certain facts or statements made by defendants:

> Shortly thereafter, an explosion is audible in the video and R repeatedly yelled "good shot my boy" and "Fuck 12." *I know from my training and experience that* the term "Fuck 12" is a derogatory phrase often directed at law enforcement officers (our italics, PC_FBI_2020).

> I spoke with Z. Z said he does use "dabs". *I know from my training and experience that* dabs is a commonly used name for hashish oil (our italics, PC_ND_MandanPD_2018).

In these two extracts, the authors use their police knowledge to explain the two terms to the lay reader(s) in order to secure the understanding of the meaning of these statements

used by defendants (a pejorative expression to refer to the police in the first example and the designation of illegal drugs in the second one).

Moreover, the authors' credibility and seriousness and their status as a bearer of truth are also enhanced by the fact that they officially take an oath and declare on their honour the truthfulness of the narrated events. PCAs are sworn statements made in writing before a competent authority (notary public, deputy clerk of the court, assistant state attorney, magistrate or certified officer). This is indicated by the words *Before me* in the sentences "Before Me, the undersigned authority, personally appeared [name of police officer] . . ." or "Subscribed and sworn to (or affirmed) before me" at the beginning or end of documents. When signing an affidavit or sworn statement, the police officer solemnly declares that the stated facts are true. This is reflected in the use of frequently used fixed phraseology such as "The undersigned certifies and swears that . . ." or "I swear that the above statement is correct and true to the best of my knowledge and belief". In the event of perjury—lying or giving false evidence—police officers are liable to severe penalties, including dismissal, redundancy or imprisonment. John Michael Callahan, deputy sheriff for Plymouth County (Massachusetts) and former NCIS and FBI special agent, outlines the consequences of deliberate misrepresentation and omission in affidavits drafted by American officers:

> Carlos Luna, a Boston Police Department (BPD) Detective, obtained a search warrant for a residence based upon his sworn affidavit. Luna's affidavit claimed he received information from an informant that illegal drug activity was occurring at that residence. Luna and other officers went to the residence to execute the warrant. During a forced entry, shots were fired from inside the residence and an officer was killed. Albert Lewin was charged with murder of the officer. During legal proceedings that followed, Lewin's lawyer moved for disclosure of Luna's confidential informant. The judge granted the motion, but the prosecution was unable to produce the informant. As a result, the trial judge dismissed the Lewin indictment. Detective Luna submitted a new affidavit in an effort to obtain reinstatement of the charges against Lewin. Luna admitted to making substantial material misstatements in his search warrant affidavit including the facts that he attributed to his informant. The case against Lewin was reinstated by the Massachusetts Supreme Judicial Court, but Lewin was later found not guilty of the officer's murder at trial. Detective Luna was subsequently charged and convicted of perjury and filing false police reports (Callahan 2019).

The authors' specialised knowledge, their position within the specialised community and the action of oath-taking are elements that guarantee and reinforce the seriousness and reliability of the facts narrated in probable cause affidavits. Additionally, the expertise and credibility of the authors also legitimise the signposting work they perform when classifying the offence, thus laying the foundations of the judicial process.

### 4.2. Signposting and Classification of the Offence

When they look at the documents in a case file, actors involved in the judicial system must be able to quickly identify the type of case presented and, in particular, the category of the offence. Therefore, the police carry out an operation of signposting, which consists of classifying the case in one (or more) specific category(ies) of criminal offence(s). This initial classification conditions the reception of the text as a whole, as it orients the case towards a defined legal framework and, consequently, towards the nature of the expected evidence. To follow the metaphor of the railroad switch on a railway, the author of a probable cause affidavit drives the case in the direction of one or more common law precedents and directs it towards legal lines of final destination that have been determined over the decades by case law: "legislators codify offences *ex ante*, and [. . .] police and prosecutors confine their collective attention to the catalogue of what has already been defined as criminal" (Bowers 2014, p. 997). By classifying the offence, American police officers attempt to insert the facts into the wider context of the legal system. In order to do so, the police specifically name the offences that were committed and refer to the corresponding legislation. This

aspect is illustrated by the use of specific legal terminology and, more precisely, fixed phraseological units both in the peritext (Figure 5, example 1) and in the body of the text (Figure 5, example 2).

PC_FL_DaytonaBeachPD_2020

PC_FL_BrevardCountySO_2020(1)

**AFFIDAVIT FOR ARREST WARRANT**

State of Florida
County of Brevard

BEFORE ME ▮▮▮▮▮▮▮▮▮▮▮▮▮▮▮ a sworn law enforcement officer, personally came Agent ▮▮▮▮▮▮▮▮▮▮, of the Brevard County Sheriff's Office, who being duly sworn deposes and says: that Affiant has reason to believe and does believe that probable cause exists for the arrest of ▮▮▮▮▮▮▮▮▮▮, **Date of Birth** ▮▮▮▮▮▮, **Social Security Number** ▮▮▮ ▮▮▮▮▮, **last known address of** ▮▮▮▮▮▮▮▮▮▮▮▮, 5'2" 160lbs, **Black Female** for a violation of the laws of the State of Florida, to wit **Solicitation to commit 1ˢᵗ degree Murder**, contrary to section **777.04 (4-B)**, Florida Statutes and **Solicitation to commit an Occupied Burglary with a Battery**, contrary to section **777.04 (4-C)** which occurred at ▮▮▮▮ ▮▮▮▮▮▮▮▮▮▮▮▮▮, **Brevard County, Florida, 32904.**

**Figure 5.** Classification of the case by the police in PCAs.

Therefore, the type of criminal offence(s) is clearly stated. It can easily be identified by a reader who is unfamiliar with the case, as the terms used by the authors reflect how offences are referred to in legal texts: *child abuse without great harm* and *aggravated assault with a deadly weapon without the intention to kill* (example 1 above), a *solicitation to commit first-degree murder* and *solicitation to commit an occupied burglary with a battery* (example 2).

As pointed out in Section 3.2, the first designation of the offence in probable cause affidavits is not always based on a precise classification because the account reflects the imprecise initial knowledge of the facts available to the police officer when assigned to the case. The investigation then enables the classification of the criminal offence more precisely, and this progression is sometimes perceptible. For example, the relatively vague reference to *a sexual offence* at the beginning of affidavit PC_FL_ClermontPD_2020 is then classified more narrowly when the police officer uses precise legal terms: *Lewd or Lascivious Exhibition in Violation of Florida State Statute 800.04 7(a)1.* Similarly, in PC_OK_RogersCountySO_2018, the first reference to the offence is *having sex with a pony*, and it then becomes *Indecent Exposure and Bestiality* because the incident is associated with a specific and defined legal framework. Therefore, good knowledge of common law precedents and legal texts is an essential prerequisite for the authors. Police officers need to be familiar with existing legal frameworks, but they also need to continually update their knowledge because the definitions given to offences in legislative texts may evolve insofar as adaptations and modifications are necessary when a particular context arises. For example, the COVID-19 pandemic led to the implementation of new legislation (lockdowns, various bans and restrictions, border closures, etc.). In this context, an individual who deliberately coughed on a shop assistant (to protest against social distancing measures) was detained for aggravated assault:

> Based on the verbal/Written statements obtained on scene, Deputy C charged C with aggravated assault, given C intentionally and unlawfully threatened, by word or act, (coughing on) to do violence to P. At the time the threat was made (during the COVID-19 pandemic), C appeared to have the ability to carry out the threat, by active coughing on P. C's threat created in the mind of P a well-founded

fear that the violence was about to take place, and assault was made either with a deadly weapon or with a fully formed conscious intent to commit a felony (PC_FL_VolusiaCountySO_2020).

Finally, the police officers' classification of the offence is not always definitive, as it may be re-classified later in the judicial process in light of the evidence provided by investigations. Therefore, the communicative aim at work in PCAs is to construct a modelled discourse that can be correctly interpreted within a given context of jurisdictional precedents. Police officers' operation of signposting is reflected not only in the initial classification of the offence but also in the progressive construction of the burden of proof.

*4.3. The Progressive Elaboration of the Burden of Proof*

In PCAs, the burden of proof is built up through the accumulation of evidence. Police officers select from the wide range of information they receive and give priority to the decisive, even incriminating elements: "Probable cause is built like a stack of blocks—by piling one fact indicating guilt on top of another" (South Carolina Law Enforcement ETV Training Program 1976a, p. 9). The combination of verbal and physical evidence reinforces the probative force of the elements presented by the author. Additionally, writers of probable cause affidavits also diversify and multiply the sources of information: statements from the victim(s) and witness(es), interviews with suspects, evidence gathered by peers (police officers, scientific experts . . .), observations made at the scene of the incident, video-surveillance, etc. The progressive elaboration of the burden of proof is illustrated by the following PCA, in which a police officer interviews the victim and collects verbal and physical evidence:

> *I asked N to explain to me what happened.* N stated that he was bagging B's groceries and B got upset because he didn't like the way he was putting his chips into the bags. N stated after the groceries were bagged and the bill was paid B started to walk away. B then turned around and approached him and stated "Do you have a problem with me, because I have a problem with you". N then thinking that B was joking with him stated "do you?". [. . .] Then B quickly moved in N's direction and grabbed N by the throat/neck area and pushed him back against the register. [. . .] N then showed me where B placed his hand around his neck/throat. *I did observe there to be a dark red area to N's neck/throat. The area did look as it was turning to bruising. I did photograph this as evidence.* [. . .] *I asked N to provide me a written statement of the incident*, which he agreed to. *This incident was caught on the store video system. I reviewed the footage and did find that B in fact did grab/strike N in the throat area and pushed him up against the register* (our italics, PC_PA_FairviewTownshipPD_2019).

This extract exemplifies the use of two discursive and rhetorical strategies from the "reality production kit" (Hepburn 2003, p. 181). Corroboration and consensus (narrative corroborated by a witness/the victim), as well as active voicing (quotations to present supporting views), are used by the author to construct their arguments. In some cases (as in the above example from PC_PA_FairviewTownshipPD_2019), the reader can easily reconstruct the dialogue with one or several interlocutor(s). However, on many occasions in the corpus, the role of the enunciator disappears in order to place the emphasis on the collected statements and their content. The different steps of the investigation then become implicit:

> J stated P came into the office with regards to questions about the property. P started talking about a football game which led to a conversation about Collin Kaepernick. Conversation became heated and P became confrontational and threatening towards J (PC_FL_PortStLuciePD_2018(1)).

In such cases, discourse is modelled so that the questions asked by the investigators disappear in order to give primacy to the statements and evidence. Some affidavits are even characterised by a disappearance of the officers' actions in order to encourage the reader to

concentrate on the description of the facts relating to a breach of the law. This rhetorical strategy leads to the production of affidavits centred almost exclusively on account of the suspect's actions during the commission of the offence. This focus is adopted by various documents, including PC_UT_LaytonPD_2019:

> On 11/7/19 a male later identified as V ordered food from McDonald's inside of Layton Wal-Mart at anonymous-address. V then left with his food. V was wearing a dark blue sweater and blue jeans. V later returned to McDonald's and went behind the front registers into the employee area where customers are not allowed. V then proceeded to assault an employee at the register with his fists hitting the employee in the face. V then walked further back in the business into the kitchen area and assaulted another employee with his fists hitting the employee in the face as well. V then is heard saying you got my order wrong. The event was captured on surveillance cameras. V was identified by another officer on the Davis Crime Bulletin (PC_UT_LaytonPD_2019).

This affidavit mainly presents the facts that occurred and the temporality of the investigation disappears in favour of the temporality of the offence. As a result, readers of the affidavit, that is to say, outsiders who were not present at the scene, cannot measure the way in which the police officer guided, or even influenced, the exchange and the type of evidence gathered (Komter 2001, p. 368).

To put it in a nutshell, the aim of PCAs is to convince the competent judicial authorities to validate the existence of probable cause. In order to do so, police officers provide a modelled narrative of the facts and of police actions. As in most of the reports they write, police officers are not required to explicitly present a subjective analysis of the facts, and the aim is to convince by recounting events and presenting them following specific discourse conventions. Therefore, when writing police reports, police officers are part of a hybrid temporality because they are looking both to the past—events that have taken place—and to the future, as the documents will then be used in the judicial process and the future reception of the text by the reader(s) needs to be taken into account.

## 5. Conclusions

To conclude, when drafting probable cause affidavits, the police must gather sufficient evidence—both qualitatively and quantitatively—to justify the existence of probable cause and, ultimately, to support the hypothesis of the respondent's guilt.

Each move of the three-act prototypical structure of PCA described in Section 3.1 is meant to serve this mechanism. Indeed, the chronological and structured narrative of events is designed to persuade readers by presenting the facts in a logical, rational and coherent way. Move 1 (presentation of the context) places the case in a specific time and place, and the triggering event exposes a problematic situation that justifies police intervention. The second move (presentation of the investigation) enables legal authorities to assess the quality and quantity of the gathered evidence. Finally, in the last rhetorical move, the author evokes the details of police actions conditioned by the existence of probable cause (the arrest or the application for a warrant) and the resolution of the case. By rationally presenting a logical sequence of events (as in a demonstration), the author uses *logos*, one of the three rhetorical modes of persuasion defined by Aristotle—along with *ethos* and *pathos*—in his work *Rhetoric* (Chiron 2007). *Logos*, or persuasion through discourse, consists in showing that something is true or appears to be true, and this is precisely the aim of police officers when they present the details of the case in a coherent, chronological and structured narrative.

Furthermore, it can be argued that the shift from probability to certainty is also reinforced by the emphasis placed on the expertise and credibility of the author. This rhetorical strategy, referred to by Aristotle as *ethos*, is related to persuasion by character and consists in making the speaker worthy of belief through discourse. PCA authors present themselves as credible experts in the field of law enforcement, taking an oath before a competent authority and putting forward the specialised knowledge they acquired through

training and experience. Last but not least, the facts stated in probable cause affidavits participate in the initial classification of the offence, which can have a long-lasting impact on the case, and the burden of proof is progressively built. Once again, contrary to what the term *probable* suggests, there is no room for probability, doubt or uncertainty in probable cause affidavits, as the narrative does not highlight the probable dimension of the narrated facts but rather posits their veracity.

The present paper intends to contribute to characterising a barely-studied specialised language—English for Police Purposes—by providing an extensive and in-depth analysis of probable cause affidavits. It sheds light on the underlying communicative goal, as well as on the rhetorical moves and strategies that define this discourse genre, thus allowing a better understanding of the linguistic conventions and practices of American police professionals. It is hoped that these findings will be of interest to practitioners but also teachers and learners of police English, as well as to researchers characterising specialised varieties of English. Several lines of enquiry regarding police discourse remain open for future research, such as detailed and comparative studies of corresponding or related documents written by law enforcement officers from other English-speaking countries or in-depth analyses of other EPP genres (both spoken and written). Police language constitutes a promising and multi-faceted object of study that remains, for the time being, a relatively uncharted research territory in the ESP community.

**Funding:** This research received no external funding.

**Institutional Review Board Statement:** The study was conducted in accordance with the Declaration of Helsinki, and approved by the Institutional Review Board and Ethics Committee of Nantes University (protocol code 06102023) on 6 October 2023.

**Data Availability Statement:** The data presented in this study (i.e., extracts from probable cause affidavits written by American police officers) are available at http://www.thesmokinggun.com/documents, (accessed on 9 January 2021).

**Conflicts of Interest:** The author declares no conflict of interest.

## Notes

[1] For a detailed typology of discursive genres in English for Police Purposes, see Cartron (2022, pp. 173–96).

[2] The term *probable cause affidavit* dominates, but it can vary depending on the police forces. Several designations have been identified: *affidavit of (or for) probable cause*, *affidavit for an arrest warrant*, *arrest affidavit*, *charging affidavit*, *complaint affidavit*, *probable cause affidavit*, *probable cause letter,* and *probable cause statement* (or *statement of probable cause*). Despite the variety of names used to designate this type of specialised text (affidavit, statement, or letter), their content and purpose remain identical.

[3] *Affidavit* is a term borrowed from the medieval Latin *affidavit*, third person singular of the perfect indicative of *affidare*, which means "to declare under oath".

[4] The presumption of innocence is based on the principle that a person is innocent until proven guilty.

[5] The *Smoking Gun* website is famous for proving, in 2008, that an article in the *Los Angeles Times* entitled "An Attack on Tupac Shakur Launched a Hip-Hop War" was based on false documents, which led the newspaper to withdraw the article and publish an official apology (Rainey 2008).

[6] The optical recognition software is available online at https://ocr.space (accessed on 8 February 2021).

[7] To efficiently analyse the collected documents and be able to easily identify the sources of studied items, a file was created for each text, and a standardised naming system was elaborated. The files were named as follows: PC[for probable cause]_[US Postal Service code for the state, for instance, LA for Louisiana]_[Police force]_[Year]. To indicate the police force, abbreviations were used, such as *PD* for a *Police Department*, *SO* for a *Sheriff's Office,* or *FBI* for the *Federal Bureau of Investigation*.

[8] Names, addresses, and personal details were redacted to follow the ethical guidelines and policy of the journal.

[9] The Concordance plot tool of AntConc shows where a search word or expression is located in the texts. The length of the text is represented by the width of the blue bar, and each hit is indicated as a vertical line within the bar.

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
