# Peer review of "A Study of a Specialised American Police Discourse Genre: Probable Cause Affidavits"

_languages, doi:10.3390/languages8040259_

Round 1
Reviewer 1 Report
This paper analyses the genre of probable cause affidavits (PCAs) and finds three moves to the organisational structure of PCAs and then goes on to describe how these moves work to convince of probable to (perhaps) certain guilt. It is an interesting and important look at a genre which the paper suggests has not yet been explored. The article is well structured and is accessible to those who are not familiar with PCAs or the context.
I would certainly recommend this paper for publication, but I was left wondering about the comparison between this genre and academic writing as opposed to other genre and texts in legal systems (and beyond). This comparison is made in a few places, firstly the reference to Swales on Page 4 line 199, “Following Swales (1990), who studies the IMRAD (Introduction, Methods, Results, Analysis, Discussion) structure of research articles” and later again at the top of page 11. Given that there are many other texts designed to be persuasive the relevance of this could be better justified. That would help the reader understand the application of academic writing structures to the findings of the sequence, organisational structure or moves of the PCAs.
So, I am less convinced of the comparison to academic writing and how that furthers our understanding of this specific genre. In comparison to say, storytelling, or how people construct arguments perhaps borrowing from the ‘tool kit’ of DA looking at how people structure narratives. I’m thinking of Hepburn’s (2003) reality production ‘tool kit’ which explains some of what the authors comment on like ‘category entitlement’, ‘corroboration and consensus’ and ‘active voicing’ common to discursive analysis.
Author Response
Modifications have been highlighted in the revised version of the article:
- in yellow for modifications related to the redacting of names, addresses and personal details to comply with the ethical guidelines and policy of the journal.
- in green for modifications suggested by the first reviewer.
- in blue for modifications suggested by the second reviewer.
Reply to Reviewer 1's report:
Thank you for your review and your comments.
- The reference to Swales on Page 4 line 199, was removed and replaced by “Following Swales’s genre analysis approach (1990),”. I believe that it is important to still mention Swales as he is one of the pioneers of the study of rhetorical features in genre analysis and it seemed important to me to refer to this theoretical framework.
- The comparison with academic genres (top of p.11) was meant to emphasize the common feature of the credible representation of authors and their work.
- Paragraphs were added at the end of p.11 and p.14-15 to include Hepburn’s (2003) reality production ‘tool kit’. Thank you for mentioning this reference, it is very interesting and in perfect adequation with this section of my analysis of PCAs.
Reviewer 2 Report
The present paper presents a qualitative corpus-based analysis of American PCAs, in which the author aim to describe the rhetorical and textual structure of this genre for EPP purposes.
While I recognise the novelty of the topic and I found the paper sincerely a very interesting read, there are several issues to fix before the study is ready for publication. I detail these in my comments below:
I was expecting a paragraph between the introduction and the methods sections in which the authors discuss previous research on PCAs in linguistics. I think it would be useful for the reader to have an idea of what else has been done in the area beside this study
line 70 and following: please indicate in which paragraph specifically you will discuss these things
line 86: "based on genres" what does this mean? in what sense?
lines 102-104: sentence that begins with "Indeed" is too long and intricate to express a relatively plain concept. Rephrase
lines 136 -138: I wouldn't say that you "came across" your data source. It makes it sound not professional and casual. Rephrase to say that you CHOSE that website as your source. It is a deliberate research choice, not a random Google search (even if you happened upon it!)
lines 155-158: sentence is too long, split into different sentences.
lines 159-160: I am curious as to why the authors decided to "correct" typos and misspellings, hence actively modifying the original texts. Usually a transcription should be as faithful to the original as possible. The authors should make this choice clearer for the reader: what was considered to be an "error"? How were "missing words" integrated?
line 168: why is this genre complex?
lines 179-180: it is not enough to say that you performed "manual analysis". Which type? How was the text analysed? By how many people? You don't talk of methodology at all.
Section 3: At the beginning of section 3 you mention Rhetorical Move Analysis which was never mentioned before. You should thoroughly describe your methodological approach and procedure in section 2, complete with interrater agreement if more than one coder was involved.
in Table 1, remove all linguistic markers that occurred only once - if you want to make the case of "recurrent" patterns. Or you could subdivide the linguistic elements you found into frequency brackets: very frequent, moderately frequent, infrequent
line 243: Again, "a study of the context" is not enough information about your (I assume) concordance analysis. Figure 2 is very unclear (I never used AntConc myself for corpus analysis, and this is not a figure I was expecting). I would remove it or provide in-depth explanation.
line 276: statistically significant how? What test was conducted and on which data?
line 329: I agree with the authors, but I suggest this low vocabulary richness also suggests that PCAs are very formulaic in nature: it is a genre that police officers are requested to write very often, and hence collocations become fixed and routinesed for priming effects.
lines 449-450: what are the characteristics of a sworn statemet? You should provide evidence for your claim. The few examples below this sentence can be seen as anecdotal, as from your text it seems that these features only appear in the two documents you mention. I think this is a very interesting point, but you should (1) find literature on the matter (if it exists) and (2) make your argument more convincing.
The last three paragraphs before the conclusions - albeit interesting - don't really go together with the remainder of the paper. Up until now, the paper provided a corpus-based rheorical move analysis of PCAS, and each move has been analysed in depth. I strongly suggest that the author "collapse" paragraphs 3.2.1, 3.2.2, and 3.2.3 into one distinct paragraph 4 named something like "additional rhetorical strategies". More specifically, I also suggest to rename paragraph 3 into something clearer like "Move Analysis".
The methodology used for the analysis in this paragraph 4 should also be made clear in paragraph 2.
line 611: cite which edition of Aristotle's Rhetoric you are referncing. Logos and Ethos - being foreign words - should be italicesed.
Conclusions: the conclusions should reference more clearly what you found and why this is relevant and important for your target audience. Why is this study innovative? What have you found that was not known before? How have you found it? What other lines of research remain open? What were the limitations (if any) of this study?
English is at a good enough standard, although I suggest minor linguistic revisions. A general re-reading of the paper with fresh eyes will be enough, in my opinion.
Author Response
Modifications have been highlighted in the revised version of the article:
- in yellow for modifications related to the redacting of names, addresses and personal details to comply with the ethical guidelines and policy of the journal.
- in green for modifications suggested by the first reviewer
- in blue for modifications suggested by the second reviewer.
Answers to the remarks of reviewer 2:
Thank you for your review and your detailed comments and suggestions.
- I was expecting a paragraph between the introduction and the methods sections in which the authors discuss previous research on PCAs in linguistics. I think it would be useful for the reader to have an idea of what else has been done in the area beside this study
Answer: To my knowledge, no linguistic analysis has been conducted on PCAs. A paragraph has been added l.70-82. about discourse analysis in English for Police Purposes (references were added to the reference list) and the lack of research on the topic under study.
- line 70 and following: please indicate in which paragraph specifically you will discuss these things
Answer: Done
- line 86: "based on genres" what does this mean? in what sense?
Answer: Modified in the text.
- lines 102-104: sentence that begins with "Indeed" is too long and intricate to express a relatively plain concept. Rephrase
Answer: Modified in the text.
- lines 136 -138: I wouldn't say that you "came across" your data source. It makes it sound not professional and casual. Rephrase to say that you CHOSE that website as your source. It is a deliberate research choice, not a random Google search (even if you happened upon it!)
Answer: More details were added on the choice of the website (l.149-153)
- lines 155-158: sentence is too long, split into different sentences.
Answer: Modified in the text.
- lines 159-160: I am curious as to why the authors decided to "correct" typos and misspellings, hence actively modifying the original texts. Usually a transcription should be as faithful to the original as possible. The authors should make this choice clearer for the reader: what was considered to be an "error"? How were "missing words" integrated?
Answer: Actually, errors occurred during conversion (from image to text) and certain words were truncated, misspelled, etc., hence not matching the original text. As a result, proofreading was done to make sure that the original text and the one obtained through the optical recognition software were identical. I rephrased the sentence to avoid misunderstanding.
- line 168: why is this genre complex?
Answer: Modified in the text.
- lines 179-180: it is not enough to say that you performed "manual analysis". Which type? How was the text analysed? By how many people? You don't talk of methodology at all.
Answer: Modified in the text.
- Section 3: At the beginning of section 3 you mention Rhetorical Move Analysis which was never mentioned before. You should thoroughly describe your methodological approach and procedure in section 2, complete with interrater agreement if more than one coder was involved.
Answer: Sections 2.1. and 2.3. were reorganized and/or completed in order to better introduce the theoretical and methodological framework of rhetorical move analysis.
- in Table 1, remove all linguistic markers that occurred only once - if you want to make the case of "recurrent" patterns. Or you could subdivide the linguistic elements you found into frequency brackets: very frequent, moderately frequent, infrequent
Answer: Modified in the text.
- line 243: Again, "a study of the context" is not enough information about your (I assume) concordance analysis. Figure 2 is very unclear (I never used AntConc myself for corpus analysis, and this is not a figure I was expecting). I would remove it or provide in-depth explanation.
Answer: Modified in the text. + A note was added to explain how to read and understand Figure 2.
- line 276: statistically significant how? What test was conducted and on which data?
Answer: Modified in the text.
- line 329: I agree with the authors, but I suggest this low vocabulary richness also suggests that PCAs are very formulaic in nature: it is a genre that police officers are requested to write very often, and hence collocations become fixed and routinesed for priming effects.
Answer: I agree with this suggestion that was added at the end of the §. Thank you.
- lines 449-450: what are the characteristics of a sworn statemet? You should provide evidence for your claim. The few examples below this sentence can be seen as anecdotal, as from your text it seems that these features only appear in the two documents you mention. I think this is a very interesting point, but you should (1) find literature on the matter (if it exists) and (2) make your argument more convincing.
Answer: Rephrased to avoid misunderstanding.
- The last three paragraphs before the conclusions - albeit interesting - don't really go together with the remainder of the paper. Up until now, the paper provided a corpus-based rheorical move analysis of PCAS, and each move has been analysed in depth. I strongly suggest that the author "collapse" paragraphs 3.2.1, 3.2.2, and 3.2.3 into one distinct paragraph 4 named something like "additional rhetorical strategies". More specifically, I also suggest to rename paragraph 3 into something clearer like "Move Analysis".
Answer: In the instructions for the author, the journal indicates that authors have to adopt a very specific and formatted structure that includes the following sections: Introduction, Methods, Results, Discussion, Conclusion. This is the reason why I used this outline. However, I completely agree with the reviewer’s remark and it would be more coherent to add a fourth section dedicated to the presentation of additional rhetorical strategies. I have changed the titles of section 3 to go in this direction. The presentation of the outline has also been modified, both in the abstract and in the introduction.
- The methodology used for the analysis in this paragraph 4 should also be made clear in paragraph 2.
Answer: Modified in the text.
- line 611: cite which edition of Aristotle's Rhetoric you are referncing. Logos and Ethos - being foreign words - should be italicesed.
Answer: Modified in the text.
- Conclusions: the conclusions should reference more clearly what you found and why this is relevant and important for your target audience. Why is this study innovative? What have you found that was not known before? How have you found it? What other lines of research remain open? What were the limitations (if any) of this study?
Answer: Paragraph added at the end of the conclusion.
Round 2
Reviewer 2 Report
My comments were addressed convincingly in the text. My opinion is that the manuscript is ready for publication